Molecular epidemiology of carbapenem-resistant Escherichia coli in a tertiary hospital located in the Dabie Mountains region, China

Wang Lu 1 2
Gong Wanxian 2
Zhang Jie 2
Zhang Runan 3
Jing Ying 3 13126692224@163.com
Xu Yuanhong 1 xyhong1964@163.com
1 Department of Clinical Laboratory, First Affiliated Hospital of Anhui Medical University , Hefei , China
2 Department of Clinical Laboratory, Lu’an People’s Hospital (Lu’an Hospital Affiliated to Anhui Medical University) , Lu’an , China
3 Department of Clinical Laboratory Center, Beijing Children’s Hospital, National Center for Children’s Health, Capital Medical University , Beijing , China
García-Contreras Rodolfo
Electronic publication date: 2025 Oct 24
Publication date: 2025
Volume: 13
Electronic Location ID: e20188
Received 2025 Mar 7; Accepted 2025 Sep 15
Copyright: © 2025 Wang et al.
Copyright year: 2025
Copyright holder: Wang et al.
License: This is an open access article distributed under the terms of the Creative Commons Attribution License, which permits unrestricted use, distribution, reproduction and adaptation in any medium and for any purpose provided that it is properly attributed. For attribution, the original author(s), title, publication source (PeerJ) and either DOI or URL of the article must be cited.
License URL: https://creativecommons.org/licenses/by/4.0/

Keywords: Carbapenemases, Escherichia coli, Next-generation sequencing, Antibiotic resistance, Virulence factors, Sequence type

Funding: Anhui Province’s Higher Education Institutions for Key Natural Science KJ2019A1256 Lu’an City Science and Technology Program 2023kykt24 This work was supported by the Anhui province’s higher education institutions for key natural science research projects (grant number KJ2019A1256) and the Lu’an City Science and Technology Program (grant number 2023kykt24). The funders had no role in study design, data collection and analysis, decision to publish, or preparation of the manuscript.

==============================
Background

The emergence of carbapenem-resistant Escherichia coli (CREc) strains poses a growing threat to global public health, presenting significant clinical and therapeutic challenges. Although extensive studies have been conducted in urban areas and high-incidence countries, data on the prevalence and molecular characteristics of CREc in rural regions remain limited, particularly in areas like the Dabie Mountains in China. This knowledge gap is critical because regional variations in resistance mechanisms may differ substantially. This study aims to address this gap by investigating the molecular epidemiology, genetic diversity, and resistance mechanisms of CREc in a tertiary hospital located in this under-researched region, providing crucial data for regional surveillance and informed public health interventions.

Patients and methods

Between 2018 and 2022, 33 CREc isolates were obtained from 33 patients at a tertiary hospital in the Dabie Mountains region of China. We performed a retrospective clinical analysis of the patients, followed by next-generation sequencing (NGS) and comprehensive bioinformatics analysis of all 33 CREc isolates. Additionally, phenotypic tests for carbapenemase and AmpC-type β-lactamase production were carried out.

Results

This study analyzed 33 CREc clinical isolates from a mountainous region hospital in China. The isolates predominantly originated from elderly patients (66.7% aged ≥ 60 years) with comorbidities (75.8%). Phenotypic analysis showed that 97.0% of isolates produced carbapenemases (n = 32), with the gene encoding New Delhi Metallo-β-lactamase (blaNDM) variants (n = 30) dominated by blaNDM-5 (n = 24) and a notable proportion of blaNDM-13 (n = 4). In addition to carbapenemase genes, the most prevalent resistance genes were those conferring resistance to sulfonamides (97.0%, 32/33) and aminoglycosides (93.9%, 31/33). Notably, 36.4% (n = 12) of isolates exhibited fosA3-mediated fosfomycin resistance, with universal co-carriage of extended-spectrum-β-lactamase (ESBL) genes. Genomic analysis identified 24 distinct sequence types (STs), with ST410 and ST692 being most prevalent. Molecular investigation localized blaNDM within diversified Tn125 derivatives and the gene encoding Klebsiella pneumoniae Carbapenemase (blaKPC-2) with truncated Tn6296 elements. Virulence factor screening detected 71 virulence genes, including highly prevalent adhesins (fimH, 84.8%) and hemolysins (hlyE, 97.0%). Plasmid profiling showed predominant IncFII (81.8%) and IncX (63.6%) replicon types.

Conclusion

This represents the first systematic investigation of CREc epidemiology in this understudied region. Our results demonstrate a high prevalence of carbapenem resistance mediated primarily by blaNDM-5, with co-occurrence of other resistance genes (fosA3) and virulence factors (fimH/hlyE/csgA).

Introduction

Carbapenem-resistant Enterobacteriaceae (CRE), particularly carbapenem-resistant Klebsiella pneumoniae and Carbapenem-resistant Escherichia coli (CREc), pose a critical threat due to limited therapeutic options and high mortality rates (Kedišaletše et al., 2023; Macareño-Castro et al., 2022; Price et al., 2022). In China, CREc has rapidly expanded geographically. National-scale surveillance by the China Antimicrobial Surveillance Network (CHINET) indicates that Escherichia coli (E. coli) accounts for 18.9% of Enterobacterales isolates, with 1.8–2.0% of these strains being CREc (Hu et al., 2020). To investigate regional trends within this national framework, we analyzed data from the Anhui Bacterial Drug Resistance Surveillance Big Data Platform (Anhui Province Health Commission, 2023). This platform aggregates anonymized data from participating hospitals throughout Anhui Province. Our analysis, based on data retrieved in May 2023, revealed a particularly alarming trend in Lu’an City—a mountainous region of Anhui Province—where CREc rates underwent a 5.75-fold increase, rising from 0.4% in 2016 to 2.7% in 2020. Carbapenems remain the first-line therapeutic option for all E. coli isolates, including multidrug-resistant (MDR) strains (Demirci, Ünlü & Tosun, 2019). However, the emergence of CREc frequently leads to treatment failure, primarily mediated by carbapenemase production, with the gene encoding New Delhi Metallo-β-lactamase (blaNDM) being the most frequently reported carbapenemase gene (Han et al., 2020; Li et al., 2021, 2024a). This resistance severely limits treatment options, necessitating the use of alternative antibiotics, including tigecycline, colistin, and fosfomycin. Among these, fosfomycin demonstrates particular promise due to its unique mechanism of action (irreversible inhibition of cell wall synthesis via MurA blockade) and retained activity against certain MDR pathogens. However, the clinical utility of fosfomycin is increasingly challenged by emerging resistance mechanisms. The predominant mechanism of fosfomycin resistance is the acquisition of fos genes, with fosA3 being the most globally widespread (Zhang et al., 2025).

Traditionally, antimicrobial resistance and virulence in E. coli were thought to be mutually exclusive. However, recent studies report a rising prevalence of hypervirulent MDR strains, including CREc, posing a critical public health challenge (Ba et al., 2024; Karbalaei et al., 2025; Loras et al., 2021; Medugu et al., 2025). Of particular concern is the emergence of virulence determinants-including adhesins (e.g., fimH), siderophores (e.g., iutA) and hemolysins (e.g., hlyA), in MDR E. coli populations. Co-expression of these virulence factors (VFs) with resistance genes enhances bacterial colonization, immune evasion, and tissue damage, resulting in worse clinical outcomes (Karbalaei et al., 2025; Shahin et al., 2019; Yousefi et al., 2023).

With the dissemination and accumulation of blaNDM-carrying CREc isolates in China, comprehensive long-term molecular epidemiological surveillance and researches have been conducted in economically developed cities or developed economies (Fu et al., 2023; Ko et al., 2023). However, data from the least developed regions and rural areas are still lacking, particularly regarding integrated analyses of VFs, sequence types (STs) in these settings. To address this gap, we set out to analyze the susceptibility of antimicrobial agents, distribution of antibiotic resistance genes, plasmid replicons, VFs, serotypes, and STs in Dabie Mountains region, China. Portions of this text were previously published as part of a preprint (Wang et al., 2023, Research Square, DOI: 10.21203/rs.3.rs-3910839/v1).

Materials and Methods

Sample collection and antimicrobial susceptibility testing

Between April 2018 and July 2022, 4,166 clinical E. coli isolates were collected from routine specimens (including, but not limited to, blood, urine, and sputum) at Lu’an People’s Hospital, which is a 2,600-bed tertiary hospital located in the Dabie Mountains of Anhui province in China. Of these, 70 isolates (1.7%) were initially classified as CREc using the VITEK 2 Compact system (bioMérieux, Marcy-l’Étoile, France). Fosfomycin susceptibilities were determined by Etest. Antimicrobial susceptibility results were interpreted following CLSI M100-S30 guidelines, except for tigecycline, colistin, and fosfomycin. Tigecycline susceptibility was evaluated using Food and Drug Administration (FDA) breakpoints (susceptible ≤ 2 mg/L; intermediate = 4 mg/L; resistant ≥ 8 mg/L). For colistin, European Committee on Antimicrobial Susceptibility Testing (EUCAST) criteria were applied (susceptible ≤ 2 mg/L and resistant > 2 mg/L). Fosfomycin results were interpreted according to EUCAST breakpoints (susceptible ≤ 32 mg/L; intermediate = 64 mg/L and resistant ≥ 128 mg/L). E. coli ATCC 25922 was used as the quality control strain. All isolates were stored at −80 °C for subsequent analysis.

Patient characteristics

After excluding duplicates, we included 33 non-redundant CREc cases (one per patient) with complete medical records. For these cases, we retrospectively collected detailed clinical data, including information from the 3 months preceding hospitalization.

Detection of carbapenem resistance phenotypes and confirmation of AmpC β-lactamases

The production of carbapenemase was assessed using the carbapenemase inhibition test (Song et al., 2024). AmpC enzymes in CREc isolates were detected using the modified three-dimensional extract test (Coudron, Moland & Thomson, 2000). E. coli ATCC 25922 was used as negative quality control.

Next-generation sequencing and phylogenomic analysis

Genomic DNA was extracted using a bacterial DNA extraction kit (Sangong Biotech, Shanghai, China). Sequencing libraries were prepared with the NEB Next Ultra DNA Library Prep Kit for Illumina (New England Biolabs, Ipswich, MA, USA). Whole-genome sequencing was performed on the Illumina NovaSeq PE150 platform. Raw sequencing data were filtered to remove low-quality reads and assembled with SPAdes.

Resistance genes, the multilocus sequence typing (MLST) STs, and plasmid replicons were identified using the Bacterial Analysis Pipeline from the Center for Genomic Epidemiology (Bortolaia et al., 2020; Carattoli et al., 2014). Bacterial VFs were identified through the virulence factor database updated in 2019 (VFDB 2019, http://www.mgc.ac.cn/VFs/) (Liu et al., 2022). Insertion sequence (IS) elements were scanned using the ISfinder database (Siguier et al., 2006). All the open reading frames or pseudogenes were predicted using the online website RAST 2.0 (Aziz et al., 2008). The mapping and comparative analysis of gene structure were performed by using BLASTN (Camacho et al., 2009) and displayed by Inkscape 1.0 (https://inkscape.org/en).

The sequences were compared with the reference sequence E. coli str. K-12 (GenBank accession number: NC_000913.3). To identify potential single nucleotide polymorphisms (SNPs), MUMmer 3.23 was used (Delcher, Salzberg & Phillippy, 2003). Tandem Repeats Finder (Benson, 1999) and RepeatMasker (Tarailo-Graovac & Chen, 2009) were used to evaluate the repeated sequence area of the reference sequence and to filter the SNPs located in the repeated area. SNP-based phylogenetic trees were built using PhyML v3.0 under HKY model (Guindon et al., 2010). Confidence was inferred by running 1,000 bootstrap replicates.

Nucleotide sequence accession number

The draft-genome sequences were submitted to GenBank under BioProject PRJNA1021959 (Table S1).

Results

Clinical characteristics, distribution, and multidrug resistance Phenotypes of 33 CREc isolates

Results showed that 66.7% (22/33) of the 33 patients with CREc infection were elderly (≥60 years old), and 45.4% (15/33) had undergone surgery. Comorbidities were present in 75.8% (25/33) of patients, all of whom underwent invasive procedures. Among these patients, 27.3% (9/33) had solid tumors, 21.2% (7/33) had hypertension, and 9.1% (3/33) had diabetes or leukemia. The most common species of the CREc isolates collected were urine (39.3%, 13/33), followed by sputum (15.2%, 5/33), and secretion (15.2%, 5/33) (Table 1).

Table 1 Clinical characteristics of 33 patients infected by CREco isolates.

Clinical characteristics	Number (%)	
Median age in years	63 (17–90)	
Age (years)		
<18	1 (3.0)	
18~45	1 (3.0)	
46~59	9 (27.3)	
60~75	17 (51.5)	
>75	5 (15.2)	
Ward distribution		
ICUa	4 (12.1)	
Surgery	15 (45.4)	
Oncology	3 (9.1)	
Nephrology	3 (9.1)	
Infectious disease	2 (6.1)	
Emergency department	2 (6.1)	
Others	4 (12.1)	
Underlying diseaseb		
Solid tumor	9 (27.3)	
Hypertension	7 (21.2)	
Diabetes mellitus	3 (9.1)	
Leukemia	3 (9.1)	
Renal disease	2 (6.1)	
Others	4 (12.1)	
No underlying medical condition	7 (21.2)	
Invasive procedureb		
Mechanical ventilation	13 (39.4)	
Drainage catheters	24 (72.7)	
Surgery	15 (45.5)	
Specimen		
Urine	13 (39.3)	
Sputum	5 (15.2)	
Secretion	5 (15.2)	
Blood	3 (9.1)	
Abdominal fluid	3 (9.1)	
Others	4 (12.1)	
Notes:

a ICU, intensive care unit.

b It should be note that the total percentage of underlying diseases and the patients with invasive procedures might exceed 100% due to the patients might have more than one underlying disease or invasive procedure.

Phenotypic testing, including the three-dimensional test and carbapenemase inhibition test, revealed that seven of the 33 CREc isolates produced AmpC enzymes, while 32 produced carbapenemases (results were shown in Fig. S1). Of 32 carbapenemase-producing CREc isolates, 30 produced metallo-β-lactamase (MBL) and the rest two produced serine β-lactamase (Table S1).

The antimicrobial susceptibility/resistance profiles of these 33 CREc isolates were determined using 19 different antibiotics (Table 2). All CREc isolates (100.0%, 33/33) were resistant to third-generation cephalosporins and at least one kind of carbapenem. Besides their high-level resistance of β-lactams, these isolates were also resistant to at least another two different categories of antibiotics, resulting in the MDR phenotype of these CREc isolates (Table S1). Notably, susceptibility rates remained high for tigecycline (100.0%), polymyxin B (97.0%), and amikacin (81.8%), but lower for fosfomycin (63.6%) (Table 2).

Table 2 Antimicrobial resistance and susceptibility rates of 33 CREco.

Antibiotic	R%	S%	
Amoxicillin-clavulanic acid	100.0	0.0	
Piperacillin-tazobactam	100.0	0.0	
Cefuroxime	100.0	0.0	
Ceftazidime	100.0	0.0	
Cefotaxime	100.0	0.0	
Cefepime	100.0	0.0	
Cefoperazone-sulbactam	100.0	0.0	
Ertapenem	100.0	0.0	
Imipenem	93.9	3.0	
Meropenem	93.9	3.0	
Ciprofloxacin	97.0	3.0	
Levofloxacin	97.0	3.0	
Trimethoprim-sulfamethoxazole	84.8	15.2	
Aztreonam	70.0	27.3	
Tobramycin	48.5	27.3	
Amikacin	15.2	81.8	
Fosfomycin	36.4	63.6	
Polymixin B	3.0	97.0	
Tigecycline	0.0	100.0	
Note:

R%: percentage of resistant isolates; S%: percentage of susceptible isolates.

Genetic characteristics and relatedness of 33 CREc

MLST analysis of seven housekeeping genes (adk-fumC-gyrB-icd-mdh-purA-recA) classified the 33 CREc isolates into 24 STs (Table S1). ST410 (n = 4) was the most prevalent, followed by ST156 (n = 3), ST692 (n = 3), ST167 (n = 2), and ST2 (n = 2) (Fig. 1). The remaining 19 isolates belonged to 19 independent STs.

Figure 1 Sankey plot showing the association between carbapenemases and STs from different sources.

ND, not detected.

Among the 33 CREc isolates, 30 (90.9%) harbored blaNDM variants, including blaNDM-5 (n = 24), blaNDM-13 (n = 4), blaNDM-1 (n = 1), and blaNDM-6 (n = 1). Notably, blaNDM-6 was found in ST361, while blaNDM-13 was found in ST744, ST1196, ST2064, and ST2179. The remaining ST206 isolate (ECO28), did not carry any known carbapenemase genes (Fig. 1). Additionally, two isolates of ST662 and ST131 carried the gene encoding Klebsiella pneumoniae Carbapenemase (blaKPC-2).

Whole-genome sequencing of 33 isolates identified 198,764 SNPs for phylogenetic analysis. Twelve of the 33 isolates showed close genetic relatedness. The first isolate (ECO01), was primarily collected in 2018, followed by five isolates in 2019 and six in 2021 (Fig. 2).

Figure 2 The genetic relatedness, resistance phenotypes and corresponding resistance mechanisms of 33 CRECo isolates.

Distribution of acquired antibiotic resistance genes (ARGs) and replicons of 33 CREc isolates

A total of 73 different acquired ARGs, involved in showing resistance to 12 categories of antibiotics, were identified in 33 CREc isolates (Table S1). The most prevalent ARGs were carbapenemase genes (97.0%, 32/33) and sulfonamide genes (97.0%, 32/33), followed by aminoglycoside resistance genes (93.9%, 31/33) (Table S1 and Fig. 2). The distribution of ARGs and their corresponding antibiotic phenotypes are shown in Fig. 2. As expected, all 32 isolates carrying blaNDM/KPC were resistant to at least two different carbapenems. Additionally, all isolates resistant to β-lactams (including amoxicillin-clavulanate, piperacillin-tazobactam, aztreonam, second-generation or higher cephalosporins, and carbapenems) exhibited resistance mechanisms co-mediated by carbapenemase genes, extended-spectrum-β-lactamase (ESBL) genes, and AmpC genes.

Interestingly, four CREc isolates carrying ESBL genes remained susceptible to aztreonam (a monobactam antibiotic), including one carrying blaOXA-10, two carrying blaCTX-M-14, and one carrying blaTEM-20. Genotypic and phenotypic resistance results were fully consistent for trimethoprim-sulfamethoxazole, fosfomycin and polymyxin. Thirty-one of the 33 isolates carried aminoglycoside-modifying enzyme (AME)-encoding genes, with four also carrying the 16S rRNA methyltransferase gene rmtB, which contributed to resistance to both amikacin and tobramycin. Eight isolates carried the acquired quinolone resistance gene qnr. Of the 33 isolates, 32 (97.0%) exhibited resistance to ciprofloxacin and levofloxacin. All 12 strains (36.4%) carrying the fosA gene were also found to co-harbor ESBL genes (e.g., blaCTX-M-14, blaCTX-M-55) (Table S1).

Seventy-one VFs were detected, with csgA, nlpI, and terC universally present (100.0%), followed by yehB, hlyE, and yehC (97.0%, respectively). Notably, over 50.0% of isolates carried the following genes:

yehD (93.9%), fdeC (93.9%), fimH (84.8%), yehA (84.8%), anr (75.8%), hha (66.7%), traT (63.6%), lpfA (63.6%), iss (63.6%), ompT (51.5%), sitA (51.5%), and traJ (51.5%) (Table S1).

Serotyping revealed 15 O types and 16 H types among the 33 isolates. The most prevalent O types were O8 (n = 6, 18.2%) and O101 (n = 6, 18.2%), while H9 predominated among H types (n = 8, 24.2%), followed by H25 (n = 6, 18.2%). A total of 21 unique serotype combinations were observed, with O8: H9 (n = 6, 18.2%) and O101: H9 (n = 4, 12.1%) representing the most common combinations.

Each of these 33 CREc isolates harbored one to seven kinds of replicons, giving a total of 13 identified replicons (Fig. S2). Among these different replicons, the most prevalent ones were FI/FII (81.8%, 27/33), and followed by IncX (63.6%, 21/33) (Table S1).

The core genetic environment of carbapenemase gene blaNDM and blaKPC in the 33 CREc isolates

Among the 33 CREc isolates, 30 carried blaNDM and two carried blaKPC-2. A detailed comparison of their genetic environments was performed (Fig. 3). The core genetic environments of blaNDM and blaKPC-2 were derived from Tn125 and Tn6296, respectively. Among the 33 isolates, seven distinct truncated ΔTn125 derivatives (ΔTn125-1 to -7) were identified, containing blaNDM-1/5/6/13 variants with the following distribution: (1) ΔTn125-1 (n = 13, ECO04/06/07/09/11/13/14/15/18/19/27/31/32), (2) ΔTn125-2 (n = 4, ECO10/12/21/30), (3) ΔTn125-3 (n = 8, ECO01/02/03/05/23/25/32/33), (4) ΔTn125-4 (n = 1, ECO20), (5) ΔTn125-5 (n = 3, ECO17/26/29), (6) ΔTn125-6 (n = 1, ECO16), and (7) ΔTn125-7 (n = 1, ECO08) (Fig. 3A); while one (ECO07) blaKPC-2-carrying isolate belonged to ΔTn6296-1 and another (ECO24) to ΔTn6296-2 (Fig. 3B).

Figure 3 (A) Comparison of Tn125 and its seven derivatives. (B) Comparison of Tn6296 and its two derivatives.

(A) Genes are denoted by arrows. Genes, mobile elements, and other features are colored based on their functional classification. Shading denotes regions of homology (nucleotide identity ≥ 95%). (B) Genes are denoted by arrows. Genes, mobile elements, and other features are colored based on their functional classification. Shading denotes regions of homology (nucleotide identity ≥ 95%).

These seven variants of Tn125 (ΔTn125-1 to ΔTn125-7) exhibited truncation at three downstream sites: cutA (periplasmic divalent cation tolerance protein), trpF (N-5′-phosphoribosyl anthranilate isomerase), and groEL (60 kDa chaperonin 2) in ΔTn125-1 to -4, ΔTn125-5/6, and ΔTn125-7, respectively (Fig. 3A); in addition, except for ΔTn125-1, the rest six Tn125 variants also displayed the truncation or insertion of the tnpA (transposase) in ISAba125.

Two variants of Tn6296, which were identified in the blaKPC-2-carrying isolates, exhibited truncations at the tnpA(transposase) and tnpR (resolvase) sites in the downstream region (Fig. 3B).

Discussion

Our investigation of CREc antimicrobial resistance patterns and virulence characteristics provides critical insights for infection control implementation. During the study period from 2018 to 2022, we observed a dynamic trend in CREc detection rates, increasing from 0.8% in 2018 to a peak of 3.8% in 2020, followed by a decline to 0.7% in 2022. Notably, the relative surge in CREc prevalence during 2020 may potentially correlate with concurrent infections during the COVID-19 pandemic (Chatterjee et al., 2023).

The CREc infections were predominantly observed in the surgery department (45.4%), particularly in urology. While previous studies have reported mortality rates as high as 30.0% in CREc infections (Kong et al., 2024), only one fatality (3.0%) occurred in our cohort, possibly reflecting the less severe disease presentation in our patients. Notably, 24.2% (8/33) of patients discontinued treatment, seven of whom were cancer patients over 50 years old, suggesting that underlying malignancies may contribute to treatment non-adherence.

blaNDM-mediated carbapenem hydrolysis represents the predominant resistance mechanism among CREc isolates (Li et al., 2021). Molecular subtyping revealed blaNDM-5 as the dominant variant (accounting for 80.0% of all blaNDM carriers), consistent with China’s predominant blaNDM epidemiology (Li et al., 2021). Secondary blaNDM subtypes included: blaNDM-13 (n = 4), blaNDM-6 (n = 1), and blaNDM-1 (n = 1). Notably, These variants (blaNDM-13 and blaNDM-6) have been infrequently reported in China (Lv et al., 2016; Wang et al., 2023; Zhang et al., 2023), highlighting the need for continuous surveillance to track their dissemination and assess their potential clinical impact. Although no carbapenemase genes were detected in ECO28, this strain showed high expression of acrA, acrB, and tolC, which may be associated with reduced susceptibility to multiple antibiotics, including β-lactams, due to the overexpression of this tripartite efflux pump system (Lian et al., 2024).

Antibiotic susceptibility testing showed that CREc isolates were most susceptible to tigecycline, polymyxin, and amikacin, aligning with the CHINET data, suggesting these antibiotics as the most effective treatments for CREc infections in China (Hu et al., 2020). We found that blaCTX-M-55 was the predominant blaCTX-M, followed by blaCTX-M-14 and blaCTX-M-15, current epidemiological data indicated that blaCTX-M-55, previously considered a rare variant, has now surpassed blaCTX-M-15 and blaCTX-M-14 in prevalence to become the predominant genotype among CTX-M-type ESBLs (Yu et al., 2024; Zhang et al., 2019, 2021). Interestingly, four CREc isolates carrying ESBL genes remained susceptible to aztreonam, likely due to these enzymes’ limited hydrolytic activity against aztreonam. Although the acquired quinolone resistance gene qnr was detected in only eight isolates, nearly all isolates (32/33) exhibited resistance to ciprofloxacin and levofloxacin, indicating that qnr genes were not the primary drivers of fluoroquinolone resistance (Strahilevitz et al., 2009). Instead, mutations in the parC gene, found in 30 of the 33 isolates, were strongly associated with high-level fluoroquinolone resistance, highlighting the critical role of type II topoisomerase mutations in mediating resistance (Boueroy et al., 2023; Garoff et al., 2018). Fosfomycin, a broad-spectrum antimicrobial agent, is indicated for the treatment of both uncomplicated and complicated urinary tract infections (Moreno-Mellado et al., 2025). Additionally, it demonstrates excellent activity against infections caused by MDR and carbapenem-resistant bacteria, particularly when used in combination therapies (Hughes et al., 2020; Sojo-Dorado et al., 2022). However, reports of fosfomycin resistance have been increasingly documented in recent years (Martínez et al., 2025; Ríos et al., 2022). In our study, 36.4% (12/33) of strains exhibited fosfomycin resistance, with 11 harboring the fosA3 gene and 1 carrying fosA4. The fosA3 gene can be located on diverse plasmid types or the bacterial chromosome, and may co-transfer with ESBL genes and carbapenemase genes via mobile genetic elements through horizontal gene transfer mechanisms (Bi et al., 2017; Zhang et al., 2025).

The 33 CREc isolates in this study belonged to 24 distinct ST types, demonstrating considerable genetic diversity. Notably, SNP analysis revealed 12 closely related isolates (SNP difference ≤ 10). Although these strains were isolated from different wards over a 3-year period (2018–2021), their genetic relatedness suggests possible: (1) persistent colonization by resistant clones in the hospital environment, (2) horizontal transfer of resistance genes via mobile genetic elements (e.g., plasmids), or (3) intermittent interpersonal transmission of dominant clones. However, the absence of environmental surveillance data and detailed patient movement records necessitates further validation of these transmission routes.

Previous global reports have identified ST167, ST410, ST617, and ST131 as common among CREc isolates (Li et al., 2024a; Ma et al., 2023). ST410 is of particular concern as it is the third most prevalent strain and has frequently been associated with blaNDM in multiple regions (Bi et al., 2017; Ko et al., 2023; Zhang et al., 2017). In our study, among the 30 CREc strains producing blaNDM, 21 different STs were identified, with ST410 being the most prevalent, followed by ST156 and ST692. Previous studies indicated that ST410 has evolved into strains with high resistance and virulence (Ba et al., 2024). For instance, ST410 strains carrying the fosA gene-by virtue of its location on a conjugative plasmid-concurrently exhibit carbapenem and fosfomycin resistance, posing a dual threat to empirical therapy and representing a critical One Health concern (Güneri et al., 2022; Peng et al., 2022). In our study, genomic analysis revealed that all four ST410 strains harbored a conserved repertoire of virulence determinants, including fimH (mediating host colonization through type 1 fimbriae), csgA (critical for curli-dependent biofilm formation), hlyE (encoding a pore-forming hemolysin), and hha (modulating invasion-related gene expression), suggesting their collective contribution to the pathogen’s adhesive, invasive, and persistence mechanisms (Jans et al., 2024; Karbalaei et al., 2025; Krall et al., 2021). Therefore, vigilant monitoring of clinical CREc transmission, along with other resistance genes, is essential.

This study showed that among the 33 E. coli isolates we analyzed, 71 unique VFs were identified, encoding siderophores (iutA, iucC, irp2, fyuA), adhesins (fimH, afa, pap, sfa), and toxins (cnf1, hly, sat). In this study, the overall frequency of virulence genes ranged from 3.0% for sat to 97.0% for fimH. The fimH-encoded type 1 fimbriae are associated with biofilm formation, while the fimH adhesin mediates bacterial proliferation and invasion-key VFs in uropathogenic E. coli (Mayer et al., 2017). irp2 and fyuA, VFs of the E. coli iron uptake system, demonstrated significant association with hypervirulence and were detected in 36.4% (12/33) of CREc isolates (Hashimoto et al., 2021; Li et al., 2024b).

Tn125 is hypothesized to have derived from the Acinetobacter chromosome due to the integration of the blaNDM gene, which was captured by two copies of ISAba125 (Poirel et al., 2012), giving rise to this ISAba125-composite transposon with the core blaNDM genetic environment (blaNDM-1-bleMBL-trpF-dsbD-cutA-groES-groEL). It was an ancestral transposon as the transporter of blaNDM and is the only blaNDM -containing transposon in Enterobacterales (Grenier et al., 2024). blaNDM and its Tn125 components (exhibiting various truncated forms) could be found within MDR regions of relevant bacterial plasmids or chromosomes (Ma et al., 2024).

In this study, seven variants of Tn125 (ΔTn125-1 to ΔTn125-7) were identified, exhibiting truncations at three downstream sites: cutA, trpF, and groEL. Tn125 frequently exists in various truncated forms (Ma et al., 2024). These ΔTn125 variants are structurally similar to those identified in multiple regions of China (Hu et al., 2022; Huo et al., 2024), suggesting that Tn125 undergoes consistent structural modifications and may follow a shared evolutionary pathway across China.

In contrast, only two Tn6296 derivatives carrying blaKPC-2 were identified in 33 CREc isolates, and their genomic organization were more conserve than the Tn125 derivatives. Tn6296, first identified in the plasmid from Klebsiella pneumoniae (Jiang et al., 2010), was generated from insertion of the local blaKPC-2 gene platform into Tn1722, leading to the truncation of mcp. The structures of two Tn6296 derivatives identified in this study were more compact due to the loss of downstream genes of Tn6376. Although such insertions or deletions in the exogenous insertion region blaKPC-2 were frequently observed in Tn6296, however the core structure surrounding blaKPC-2 (ISKpn27-blaKPC-2-ISKpn6) remained intact in Tn6296 variants, indicating ISKpn27-blaKPC-2-ISKpn6 may be the key functional unit responsible for carbapenem resistance (Song et al., 2024).

Among the 30 blaNDM-carrying isolates, four harbored the rare blaNDM-13 variant: ECO17 (ST744), ECO20 (ST1196), ECO26 (ST2064), and ECO29 (ST2179). One additional isolate (ECO22, ST361) carried the blaNDM-6 variant. Compared to blaNDM-1, blaNDM-13 exhibits two amino acid substitutions, D95N and M154L, which significantly enhance its hydrolytic activity against cefotaxime (Shrestha et al., 2015). Previous reports have identified multiple copies of blaNDM-13 within a single plasmid or chromosome, often associated with mobile genetic elements like IS30 or ISAba125, which may be linked to the transmission or stability of blaNDM-13 (Huang et al., 2022; Lv et al., 2016). In this study, some of the strains carried a truncated tnpA gene and lacked genes such as dsbD, cutA, groES, and groEL. However, other studies have reported that these genes were intact in the genomes of the blaNDM-13-carrying strains (Kim et al., 2019; Shrestha et al., 2015). This suggests that genetic rearrangements or the presence of mobile genetic elements may contribute to the variability in the genetic context of blaNDM-13. The A233V substitution in blaNDM-6 enhances its enzymatic fitness in the periplasm under Zn(II) starvation conditions compared to blaNDM-1 (Sychantha et al., 2021). The presence of the ISAba125 family transposon upstream of blaNDM-6 is consistent with previous studies, which documented that the promoter of blaNDM is partially located within the inverted repeat upstream of ISAba125 (Poirel et al., 2012). Additionally, the genetic environment of blaNDM-6 includes truncated tnpA transposase, as well as bleMBL, trpF, dsbD, and a truncated cutA. Although tnpA is truncated, some studies have shown that this does not affect the plasmid’s ability to transfer between bacteria, allowing the blaNDM-6 gene to be horizontally transferred via plasmids (Bahramian et al., 2019; Parvez et al., 2019).

This study has several limitations that should be taken into account. Firstly, the use of long-read sequencing limited the thorough analysis of rare variants of the blaNDM gene, especially blaNDM-6 and blaNDM-13, which were not included in this study. Future research should explore the environmental structure and plasmid replicons to further understand these variants. Additionally, case-control studies should be incorporated to assess the potential risk factors associated with CREc infections. It is important to note that this was a single-center study with a relatively small sample size. Therefore, prospective, multicenter, large-scale clinical trials are essential to validate and extend our findings.

Conclusions

This study provides the first dataset on molecular epidemiology and genetic characteristics of CREc isolates from a tertiary hospital in the Dabie Mountains region, China, addressing the previously unmet need for CREc profiling in the least developed regions. The presence of blaNDM-5 is the primary cause of carbapenem resistance in CREc. Notably, most of these strains also carry resistance genes to other antibiotics (e.g., fosA3) as well as various VFs (e.g., fimH/hlyE/csgA). These CREc-carrying multidrug resistance and virulence genes still display the diversified phylogenetic history in this area. Therefore, ongoing surveillance needs to be performed to prevent the further clonal expression of these CREc.

Supplemental Information

Supplemental Information 1 (A) Three-dimensional extract test results for example (B) Results of the carbapenemase inhibition test for example.

(A) Enhanced growth of the surface organism (E. coil ATCC 25922), was seen near agar slits (arrows) that contain extracts of ECO02 and ECO08 test isolates, both of which are AmpC producer; (B) The difference in zone size in the presence and absence of EDTA was ≥5mm for IPM, suggesting MBL production. EDTA: ethylenediaminetetraacetic acid; PBA: phenyl boronic acid; IPM: imipenem; MBL: metallo-β-lactamase.

Supplemental Information 2 Background information of 33 carbapenem-resistant Escherichia coli isolates.

Supplemental Information 3 The diagram of replicon types of the 33 strains.

We would like to thank Xinhua Luo for his insightful comments and suggestions on the manuscript.

Additional Information and Declarations

Competing Interests

The authors declare that they have no competing interests.

Author Contributions

Lu Wang conceived and designed the experiments, analyzed the data, prepared figures and/or tables, and approved the final draft.

Wanxian Gong performed the experiments, analyzed the data, authored or reviewed drafts of the article, and approved the final draft.

Jie Zhang performed the experiments, analyzed the data, prepared figures and/or tables, and approved the final draft.

Runan Zhang analyzed the data, prepared figures and/or tables, and approved the final draft.

Ying Jing conceived and designed the experiments, prepared figures and/or tables, authored or reviewed drafts of the article, and approved the final draft.

Yuanhong Xu conceived and designed the experiments, authored or reviewed drafts of the article, and approved the final draft.

Human Ethics

The following information was supplied relating to ethical approvals (i.e., approving body and any reference numbers):

Ethical approval was approved by the Ethics Committee of Lu’an People’s Hospital (Number: 2021LLKS006)

Ethics

The following information was supplied relating to ethical approvals (i.e., approving body and any reference numbers):

Lu’an People’s Hospital

DNA Deposition

The following information was supplied regarding the deposition of DNA sequences:

The draft-genome sequences are available at GenBank: BioProject PRJNA1021959.

Data Availability

The following information was supplied regarding data availability:

The data is available at NCBI GEO: SAMN37576063, SAMN37576064, SAMN37576065, SAMN37576066, SAMN37576067, SAMN37576068, SAMN37576069, SAMN37576070, SAMN37576071, SAMN37576072, SAMN37576073, SAMN37576074, SAMN37576075, SAMN37576076, SAMN37576077, SAMN37576078, SAMN37576079, SAMN37576080, SAMN37576081, SAMN37576082, SAMN37576083, SAMN37576084, SAMN37576085, SAMN37576086, SAMN37576087, SAMN37576088, SAMN37576089, SAMN37576090, SAMN37576091, SAMN37576092, SAMN37576093, SAMN37576094, SAMN37576095.

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
