# Peer review of "Molecular epidemiology of carbapenem-resistant Escherichia coli in a tertiary hospital located in the Dabie Mountains region, China"

_PeerJ, doi:10.7717/peerj.20188_

## Round 0.1 · original submission · Major Revisions

Please reply all reviewers comments.

Reviewer 1 ·

Basic reporting

This research explored the presence and genetic features of carbapenem-resistant E. coli obtained from patient samples within a single hospital in China. The study offers specific genetic information and details on how the isolates resist antimicrobials. However, there are several limitations, such as the limited number of samples, which impacts the data's reliability; long-read sequencing yielding imprecise information about variations in the blaNMD gene; and a lack of clear justification for the selection of bacterial isolates. Furthermore, much of the data presented is descriptive without supporting statistical analysis. Most of the study's findings consist of descriptive data lacking statistical analysis. More in-depth observations can be found in the detailed comments provided below.

Experimental design

- The study's conclusions may be biased due to the relatively small sample size of 33 patient isolates.
- The study lacks statistical analysis to determine the relationship between the demographic information of the hospital and the presence of the identified resistance genes.
- The materials and methods should clearly state the criteria used for selecting the sources of the samples included in the study.

Validity of the findings

- The introduction should explain why the study specifically focuses on Escherichia coli when hospital surveillance for carbapenem resistance typically targets a broader range of bacteria.
- On line 76, the term "animal" should be revised to "warm-blooded animal" for greater accuracy regarding the bacterium's host range.
- The sentence on line 85 regarding prevalence is unclear about the specific sources of the bacteria or samples analyzed.
- The introduction needs to emphasize the clinical importance of carbapenem antibiotics.
- The abbreviation "CREco" is not standard and should be avoided in favor of more widely recognized alternatives like "CRE-Ec" or "CP-Ec" for broader communication.
- The discussion should address carbapenem infections. Given the unclear selection criteria for isolates, the conclusion that carbapenem resistance mainly originates from the surgery room is uncertain.
- A major limitation of the study is the need for deeper analysis of blaNMD variants using long-read sequencing.
- The study's primary focus on ST410 should be discussed in relation to previous research findings.
- The discussion should include important prevention and control strategies based on the study's findings to reduce the occurrence of carbapenem-resistant bacteria.

Additional comments

- The keywords on line 73 should be listed in alphabetical order.
- The formatting (italics) of resistance genes and bacterial species names should be checked for consistency.
- Inconsistencies in the number of decimal places used for reporting percentages were observed and should be standardized.
- The abbreviation "CREC" on line 250 needs to be defined.
- The formatting of the references needs to be reviewed and corrected.

Reviewer 2 ·

Basic reporting

ABSTRACT

In Line 67, it is mentioned that ‘transposons were conserved’. Shouldn’t it be ‘are conserved?

In Lines 69 - 70, you have mentioned that you have taken CREco isolates in the Dabie mountain regions of China. Shouldn’t it be ‘from the Dabie Mountains’?
Also, is it necessary to mention the location here? It’s already mentioned in Lines 48 and 55.

Line 73 - Keywords can be improved.


INTRODUCTION

Line 77 - The word ‘empirical’ doesn't quite fit here.
Shouldn’t it be indiscriminate medication use of Third-generation antibiotics?

Line 79 - Shouldn’t it be has led instead of leading?

Line 80 - The phrase ‘in the clinic’ is not needed. Also, instead of ‘and finally posed’, shouldn’t it be ‘thus posing’? Kindly rephrase it.

Line 83 - Can you change ‘as well as in China’ to include China?
2021 monitor results?? Can you find an alternative word for monitor?
Perhaps surveillance?

Line 86 - Shouldn’t it be a mechanism of carbapenem resistance in CREco?

Line 87 - Shouldn’t it be carbapenemase genes in?

Line 88 - Can you check the spelling of characteristics?
Also, can you rephrase the sentence ‘it has been …….. was blaNDM’ to ‘The frequently reported carbapenemase gene is blaNDM.’

Line 94 - economically developed cities? Can you change it to economically developed cities or developed economies?
Also, change the less developed region to the least developed region?

Line 95 - Change it to rural regions.
‘were still lacking’ can be changed to ‘are still lacking.’

Line 100 - You have mentioned environmental factors? You didn’t address this aspect in the study. Why did u mention that?
How can you say ‘would contribute’ for sure? Can you rephrase the sentence?

Line 101 - By designation, did you mean identification of a disease?

The last sentence in the Introduction could be modified a little.

The Introduction could be more detailed. It’s lacking important information. Also, you have spoken only about NDM. What about other genes, virulence factors, and plasmids? These topics are not touched upon. The authors could update the recent literature (2025).

Experimental design

MATERIALS & METHODS

Line 103 - Can you change the subheading ‘Bacterial isolates and patients’?
Suggestions: Sampling, Setting, or anything of the authors’ choice.

Line 106 - Shouldn’t it be - which is a 2600-bed tertiary hospital?

Line 107 - kindly change the following:
Clinical data for each patient from the past three months.

Line 110 - Can you provide the abbreviation MALDI-TOF in brackets?

Lines 115 - 117 - Can you provide the abbreviations FDA and EUCAST in brackets?

Line 120 - Can you replace determination with detection?

Line 122 - Synthesis of carbapenemase or carbapenems? Check.

Line 126 - Change: extracted by using the

Line 129 - Change: sequencing data was.
‘Filtered for low-quality data’ - Change it to either ‘filtered to remove low-quality data’ or ‘screened for low-quality data’.

Line 130 - Change: then the sequencing data is assembled …..

Line 131 - MLST is a tool used to determine Sequence Type (ST). The sentence can be changed to resistance genes, the multilocus sequence typing (MLST) Sequence type (ST), and plasmid replicons.

Line 135 - Change: performed by using BLASTN.

Lines 137 - 139 - can you split this into 2 sentences? Add a full stop after the bracket. Then start the new sentence - ‘ To identify potential Single Nucleotide Polymorphisms (SNPs), using MUMmer 3.23 MUMmer 3.23 was used.

Line 141 - ‘Finally ……. Was obtained.’ - This line does not fit here. Check.

Line 145 - Check if the BioProject Number is correct.

Line 148 - was granted, not approved?

Line 150 - After medical records, add a full stop instead of a comma.

Line 151 - strike the word anonymously. Not needed.
The last sentence, ‘The institution granted …..’, is repeated. Remove it.

Validity of the findings

Lines 156 - 157 - Rephrase it. ‘ Upon investigation, it was found that ….’
RESULTS

Line 160 - Change: The most common species of the CREco isolates collected were

Line 164 - Change: Results were

Line 169 - Change: at least one kind of carbapenem.

Line 170 - Intensive resistance?? Do you mean increased resistance?

Lines 172 - 173 - Change: CREco isolates still show
Three kinds of antibiotics

Line 177 - the gene names should be in Italics. Make sure you apply this throughout the manuscript.

Change: As Figure 1 shown, the distribution of As shown in Figure 1, the 24 STs was dispersed according to…..

Line 179 - Change: from with 21 different STs. Also, can you check if it’s 30 or 23 ST? If it’s 23 STs, check the percentage

Line 191 - Change: involved in showing resistance
11 categories of antibiotics

Line 192 - CREco isolates. The second c should be in small letters. (Apply this in Line 201 also)

Lines 203 - 204: Can you rephrase this sentence? Can’t understand it.
Also, check if the antibiotic names are written properly. For instance, here you have given sulfonamide/trimethoprim, isn't the antibiotic name trimethoprim-sulfamethoxazole?

Line 207 - The word however is not required here. Remove it.

Lines 209 - 211 - Change: giving a total of 13 identified replicons.
Add: the most prevalent ones were FI/FII……… and followed by IncX…..

Line 215 - genetic contexts? Can you find an alternative word?

Lines 216 - 222 - Can you rephrase this sentence ? Put the isolate names clearly, and the numbers should be in brackets instead of the isolate name.

You have sequenced 33 isolates and obtained 24 STs. This is mentioned in the Excel file and not in the manuscript. The ST types should be discussed.


DISCUSSION

Are the first 2 paragraphs required? It describes the patients which is already mentioned in the supplementary file.
You can speak more about the ST type genes and plasmids.
Virulence genes and Plasmid replicons were identified. Why hasn't this been discussed anywhere?

Line 235 - If it’s only one patient, shouldn't it be 0.09%?

Lines 248 - 249 - You have mentioned 23 different STs. In line 179, it’s given as 21st. Can you check?
ST 167 or ST 156? Check.

Lines 252 - 257 - Remove this paragraph. This is repeated below.

Line 259 - 3 isolates? In line 61, it’s mentioned as 4. Check.

Lines 268 - 269 - ‘These findings …….. Infections in China. - Remove this sentence. It's repeated.

Line 276 - Remove from - ‘ as has been …… till line 277’. Not relevant.

Line 286 - Genetic platform or gene platform? Check.

Line 287 - 288 - Can you rephrase? The message was not conveyed properly.

Lines 290 - 291 - check the sentence, punctuation errors.

Line 295 - The references 40 and 41 are a repetition of 30 and 31. Change it.

Line 297 and 305 - check if the key genes are correct. Compare it with the results.

Line 298 - retained complete forms? Can you check this?

Line 302 - transposase or transposon?


CONCLUSION

It should be more elaborate. None of the important topics are addressed here. The giveaway of the study is not mentioned.

Line 317 - This study explores molecular epidemiology and genetic characteristics. Kindly include that.
Add: ‘.....CREco isolates from a tertiary hospital in the Dabie….’

Line 320 - add or adds? Check.

Line 324 - Name should be in Caps.

Additional comments

The fosA3 genes were present in most of the isolates. The authors have not discussed this fosfomycin resistance. It would be more interesting to see the phenotypic data (MIC data) for fosfomycin resistance since most of the isolates carried the fosfomycin gene (fosA3).

The isolates carried fosA3 genes (In the Excel sheet), but no discussion is available regarding that because Nitrofurantoin and fosfomycin are the only drugs available for treatment if the rest of the drugs fail in the clinical management.

In particular, ST410 carrying fosA3 is alarming and may give a lot of new information. In the supplementary file, we can see a lot of plasmids present in these isolates.
The authors have not discussed these diverse plasmids circulating in these isolates. The plasmids may carry many AMR genes, which may be pivotal in AMR dissemination.

The authors failed to discuss the crucial virulence genes present in these isolates.
All these points could be discussed very well in the discussion.

The authors have given all the data pertaining to the study in the Excel sheet.

The Bioproject number has to be mentioned correctly in the running text.

The discussion is very weak in the manuscript.

The authors have tried to discuss the carbapenem, but even in that instance, the cited articles are very old and have to be updated in the global context.

The authors have collected the data from the hospital and done the sequencing, and a more intense global literature survey is definitely required for the study, and a better conclusion is required for accepting this manuscript in its current stage.

·

Basic reporting

The manuscript reports and discusses a critical topic of carbapenemase-resistant E. coli. The manuscript is well-written, and the study methodology and outcomes are well-characterised, but there are some minor points to be considered for improvement:
1- Overall, the manuscript requires minimal review of the English language and style.
2- E. coli must be written in italics even in the reference list.
3- More references to carbapenem-resistant E. coli should be included in the discussion section; please review the literature for relevant articles.
4- The objective of the research was to investigate molecular epidemiology; however, the total number of E. coli and/or all infections recorded at the hospital was not specified, resulting in the absence of rates or frequencies. I think it is not a significant issue, as 33 isolates were recorded from April 2018 to July 2022.

Experimental design

1- Methods should be described with sufficient detail & information to replicate.
2- The collection of bacterial isolates should be rewritten with more details and clarifications.
3. Vitek was used for antimicrobial susceptibility testing, although automated methods are not preferred for research, but are usually helpful in clinical situations.

Validity of the findings

1- Please add more clarifications to the figures to make them more informative.

---

## Round 0.2 · Minor Revisions

Please address the last minor comments.

Reviewer 1 ·

Basic reporting

Some minor errors have been detected, such as the space, the full stop, the consistency of using symbols, etc. Before publishing this manuscript, the revision should be made.

Experimental design

None

Validity of the findings

None

Additional comments

- Some references should be revised both in the text and in the reference section.
- The reference of the company of media, reagent, and tools should be addressed as (company, city, country)

Annotated reviews are not available for download in order to protect the identity of reviewers who chose to remain anonymous.

---

## Round 0.3 · accepted · Accept

Thank you for addressing the comments!

Reviewer 1 ·

Basic reporting

NA

Experimental design

NA

Validity of the findings

NA

Additional comments

NA